# Difficult-to-Treat *Pseudomonas aeruginosa* Infections in Critically Ill Patients: A Comprehensive Review and Treatment Proposal

**DOI:** 10.3390/antibiotics14020178

**Published:** 2025-02-11

**Authors:** Pablo Vidal-Cortés, Sandra Campos-Fernández, Elena Cuenca-Fito, Lorena del Río-Carbajo, Paula Fernández-Ugidos, Víctor J. López-Ciudad, Jorge Nieto-del Olmo, Ana Rodríguez-Vázquez, Ana I. Tizón-Varela

**Affiliations:** 1Intensive Care Unit, Complexo Hospitalario Universitario de Ourense, 32003 Ourense, Spain; sandra.campos.fernandez@sergas.es (S.C.-F.); elena.cuenca.fito@sergas.es (E.C.-F.); lorena.del.rio.carbajo@sergas.es (L.d.R.-C.); paula.fernandez.ugidos@sergas.es (P.F.-U.); victor.jose.lopez.ciudad@sergas.es (V.J.L.-C.); jorge.nieto.del.olmo@sergas.es (J.N.-d.O.); ana.isabel.tizon.varela@sergas.es (A.I.T.-V.); 2Hospital Pharmacy, Complexo Hospitalario Universitario de Ourense, 32003 Ourense, Spain; ana.rodriguez.vazquez2@sergas.es

**Keywords:** *Pseudomonas aeruginosa*, critically ill patient, ceftolozane–tazobactam, ceftazidime–avibactam, imipenem–relebactam, cefiderocol, ventilator-associated pneumonia

## Abstract

The management of infections caused by difficult-to-treat *Pseudomonas aeruginosa* in critically ill patients poses a significant challenge. Optimal antibiotic therapy is crucial for patient prognosis, yet the numerous resistance mechanisms of *P. aeruginosa*, which may even combine, complicate the selection of an appropriate antibiotic. In this review, we examine the epidemiology, resistance mechanisms, risk factors, and available and future therapeutic options, as well as strategies for treatment optimization. Finally, we propose a treatment algorithm to facilitate decision making based on the resistance patterns specific to each Intensive Care Unit.

## 1. Introduction

*Pseudomonas aeruginosa* is a non-fermenting, aerobic Gram-negative bacilli (GNB), first isolated in 1882. It is ubiquitous in the environment, with a particular affinity for water. *P. aeruginosa* is considered an opportunistic bacterium and has been classically associated with patients suffering from chronic structural lung diseases, burns, and neutropenia [1]. It is not part of the normal microbiota in healthy humans [2,3]; however, under conditions of high exposure pressure, such as in an Intensive Care Unit (ICU), colonization can occur in as little as 3–5 days [3].

In a 2017 World Health Organization (WHO) expert meeting held in Geneva, *P. aeruginosa* resistant to carbapenems was classified as a top priority pathogen for research [4]. This classification was updated in 2024, maintaining carbapenem-resistant *P. aeruginosa* (CR-PA) in the high-priority group [5]. The importance of *P. aeruginosa* lies in two main factors. Firstly, it is one of the leading causes of healthcare-associated infections (HAIs) [6,7], particularly in critically ill patients [8,9]. Secondly, *P. aeruginosa* exhibits intrinsic and acquired resistance to several first-line antibiotics, making treatment complex and challenging.

In this review, we summarize the epidemiology, resistance mechanisms, and therapeutic options for DTR *P. aeruginosa* in critically ill patients.

## 2. Epidemiology

### 2.1. Healthcare-Associated Infections (HAIs)

Globally, *P. aeruginosa* is the third most frequently isolated GNB among patients hospitalized in Intensive Care Units (ICUs) [8].

#### 2.1.1. Nosocomial Pneumonia

*P. aeruginosa* is, along with *S. aureus*, the leading cause of bacterial pneumonia among hospitalized patients, being the most frequently identified GNB in nosocomial pneumonia. A U.S. study reported *P. aeruginosa* in 11% of all nosocomial pneumonia cases. Ventilator-associated pneumonia (VAP) is the most severe HAI acquired in ICUs. Data from the SMART registry, comprising 7171 respiratory samples from 209 ICUs across 56 countries, identified *P. aeruginosa* in 25% of VAP cases [6,10,11,12,13].

A notable scenario involves the SARS-CoV-2 pandemic, where patients requiring mechanical ventilation (MV) exhibited a high risk of VAP. In numerous published series, *P. aeruginosa* was identified as the primary etiologic agent [14,15].

Geographic differences influence HAI etiology. In the U.S., *P. aeruginosa* ranks second as a cause of VAP [7], while it holds the top position in Europe [16] and Spain, where it accounts for 20.7% of VAP cases according to the National Nosocomial Infection Surveillance Study (ENVIN) [9].

#### 2.1.2. Other Healthcare-Associated Infections

While respiratory infections are the primary focus for *P. aeruginosa*, it is also frequently implicated in other sites, such as catheter-associated urinary tract infections (CAUTIs) [6,17], catheter-related bloodstream infections (CRBSIs), and surgical site infections (SSIs) [18].

The EUROBACT-2 study reported *P. aeruginosa* as responsible for 14.3% of CRBSIs in ICU patients, following *Klebsiella* spp., *Acinetobacter* spp., and *E. coli* [19]. Additionally, it caused 4% of CRBSIs in critically ill COVID-19 patients [20]. *P. aeruginosa* is the fourth leading cause of CRBSIs in Europe, the fifth in Spain, and the eleventh in the U.S. [7,9,16].

For CAUTIs, *P. aeruginosa* ranks as the third most common cause in the U.S. and Europe [7,16] and the second in Spain [9]. It is also the fifth most common pathogen responsible for SSIs in the U.S. [7].

### 2.2. Community-Acquired Infections

Although primarily a nosocomial pathogen, *P. aeruginosa* has been implicated in community-acquired infections, including folliculitis and other skin and soft tissue infections (SSTIs), especially those involving water exposure. It has also been associated with osteomyelitis and endocarditis in intravenous drug users and peritonitis in patients undergoing peritoneal dialysis. *P. aeruginosa* accounts for 1% of community-acquired bloodstream infections [21] and up to 5% of severe community-acquired pneumonia (CAP) [22,23].

### 2.3. Patients at Risk

Until 1980, *P. aeruginosa* was responsible for nearly half of infections in neutropenic and burn patients. Although its incidence in these populations has declined, it remains a significant cause of bacteremia [24], likely due to the use of empiric treatments with antipseudomonal activity.

A similar trend is observed in cystic fibrosis (CF) patients. The incidence of *P. aeruginosa* infections has decreased in recent years, likely due to improved acute infection treatments and prophylaxis [25]. However, *P. aeruginosa* remains the leading cause of respiratory infections in CF [26,27,28]. Data from the European CF Society registry indicate a prevalence of 30%, with half of CF patients over 40 years old presenting at least one positive culture for *P. aeruginosa* [29]. Once *P. aeruginosa* colonizes the lungs, eradication is nearly impossible [28].

A similar phenomenon occurs in patients with non-CF bronchiectasis, where *P. aeruginosa* is associated with worsening functional status and increased mortality [6,30]. *P. aeruginosa* exhibits reduced virulence in these chronic infections, but resistance becomes a significant concern [31].

Lastly, *P. aeruginosa*’s ability to persist on various materials and surfaces, especially in water-rich environments, has led to hospital outbreaks linked to inadequate medical equipment disinfection or cross-contamination of surfaces and water sources [32]. Some outbreaks have reported mortality rates of up to 30% [33].

## 3. Antimicrobial Resistance

In addition to possessing numerous virulence factors and a remarkable ability to survive in adverse environments, this organism has another critical feature that significantly complicates its treatment: resistance to multiple antimicrobials. It is intrinsically resistant to multiple antibiotics and has a high capacity to acquire new resistance mechanisms [34,35,36]. Over 40 chromosomal genes are implicated in its resistance mechanisms [37], including 8.4% regulatory genes [34], which enable it to mutate and adapt to its environment. The coexistence of combined resistance mechanisms is frequently observed [38,39].

### 3.1. Antimicrobial Resistance Mechanisms

*Intrinsic* resistance arises from the expression of efflux pumps that expel antibiotics from the cell, the production of enzymes that inactivate or hydrolyze antibiotics (e.g., extended-spectrum beta-lactamases, ESBLs, AmpC beta-lactamases, or carbapenemases), and the low permeability of its outer membrane. Particularly significant is the downregulation of the outer membrane protein OprD, which decreases membrane permeability to certain antibiotics and is often responsible for carbapenem resistance.

The inducible expression of chromosomal AmpC beta-lactamases is primarily responsible for *P. aeruginosa’s* natural resistance to most penicillins and cephalosporins [3,31,40]. Constitutive production of the MexAB-OprM efflux pump contributes to reduced susceptibility to beta-lactams, while the inducible production of MexXY partially accounts for decreased susceptibility to aminoglycosides [3].

*Acquired* resistance can occur through horizontal gene transfer (via plasmids, transposons, integrons, and prophages, from the same or different bacterial species) or mutational changes. These can confer advantages such as reduced antimicrobial uptake, modification of the antibiotic target, efflux pump overexpression, or the emergence of antibiotic-inactivating enzymes.

*Adaptive* resistance enhances bacterial survival in response to environmental stimuli and is reversible when the stimulus ceases. In *P. aeruginosa*, adaptive resistance often involves biofilm formation [41,42].

Based on consensus definitions by the European Centre for Disease Prevention and Control (ECDC) and the Centers for Disease Control and Prevention (CDC) [43], *P. aeruginosa* can be categorized as multidrug resistant (MDR-PA) if it is non-susceptible to at least one antimicrobial agent in three or more classes active against *P. aeruginosa*. Extensively drug-resistant *P. aeruginosa* (XDR-PA) is defined as non-susceptible to all but two or fewer antimicrobial classes, while pan-drug-resistant *P. aeruginosa* (PDR-PA) is non-susceptible to all antimicrobial agents. The term “non-susceptible” is used, despite its less reader-friendly nature, as it encompasses both resistant and susceptible with increased exposure concepts.

The concept of difficult-to-treat resistance *P. aeruginosa* (DTR-PA) has also been proposed, including strains non-susceptible to piperacillin–tazobactam, ceftazidime, cefepime, aztreonam, meropenem, imipenem–cilastatin, ciprofloxacin, and levofloxacin [44,45] (Table 1).

MDR-PA or DTR-PA usually result from the combination of multiple resistance mechanisms. Resistance to carbapenems often arises from the interplay of several mechanisms rather than carbapenemase production alone; however, carbapenemase production accounts for up to 20% of CR-PA cases [45], with metallo-beta-lactamases (MBLs) being the most prevalent carbapenemase group globally in *P. aeruginosa* [46,47].

### 3.2. Antimicrobial Resistance Prevalence

In 2020, the International Nosocomial Infection Control Consortium reported resistance rates of *P. aeruginosa* to major antibiotics in device-associated infections. In VAP, resistance rates to carbapenems, quinolones, piperacillin–tazobactam, and cefepime were 39.4%, 34.6%, 39.2%, and 40.5%, respectively. For CAUTIs, rates reached 39.3%, 40.2%, 38.2%, and 48.1% in the same order, while BSIs showed rates of 43.5%, 20%, 33%, and 41.7%. Amikacin was the only antibiotic with resistance rates below 30%: 24.7% in VAP, 26.8% in CAUTIs, and 21.4% in BSIs [48].

Fortunately, in recent years new antibiotics with activity against *P. aeruginosa* have been marketed.

European studies indicate that 95.5% of *P. aeruginosa* isolates are susceptible to ceftazidime–avibactam (CAZ-AVI), 94.3% to imipenem–relebactam (IMI-REL), 93.3% to ceftolozane–tazobactam (CFT-TAZ), and 88.7% to meropenem–vaborbactam (MER-VAB) [49]. Similarly, 73.4% and 71.0% of XDR-PA isolates remain susceptible to CFT-TAZ and CAZ-AVI, respectively. Resistance to both antibiotics was observed in only 3.9% of isolates, mainly due to carbapenemase production but also mutant AmpC presence [47,50,51,52,53].

A study of strains isolated between 2013 and 2018 across 24 European countries showed that 99.7% of *P. aeruginosa* isolates were susceptible to cefiderocol, including 97.5% of carbapenem-resistant isolates. In this study, only 43.2% and 44.1% of CR-PA maintained susceptibility to CAZ-AVI and CFT-TAZ, respectively [54]. Similar findings were observed in another study analyzing CR-PA isolates from 44 Spanish hospitals. Up to 30.6% were carbapenemase producers, primarily MBL of the VIM type, and 26.7% were susceptible only to colistin [55].

Studies also highlight that AmpC-AmpR region mutations associated with resistance to CFT-TAZ and CAZ-AVI during treatment reduce cefiderocol activity but seem to enhance IMI-REL susceptibility [56].

Patterns of resistance vary geographically and over time. For example, in Asia, almost 40% of isolates are MDR or carbapenem resistant, while in the U.S., resistance rates to cephalosporins and carbapenems range from 22 to 28%, with approximately 20% of isolates being MDR-PA [12,57,58,59,60,61]. A study in Spain from 2022 reported a decrease in XDR-PA prevalence from 15% to 6% over five years, with an even sharper decline in ICU settings [62]. This favorable trend has also been described in other regions of the world [16,45,60].

These findings underscore the importance of ongoing surveillance, the knowledge of local microbiological profiles, and tailored therapeutic strategies to address the challenge of resistant *P. aeruginosa*.

## 4. Risk Factors

Classic risk factors for *P. aeruginosa* infection include advanced age, structural lung diseases (e.g., cystic fibrosis, bronchiectasis), hematologic malignancies (particularly those associated with neutropenia), solid organ transplantation, extensive burns, prior antibiotic treatment within 90 days, the presence of a central venous catheter (CVC) or urinary catheter (UC), renal replacement therapies, prolonged hospitalization, and mechanical ventilation. A respiratory focus of infection and presentation with septic shock are also associated with a higher risk of *P. aeruginosa*, as is prior colonization [6,34,38].

Additional factors increase the risk of *P. aeruginosa* as a causative agent of community-acquired pneumonia (CAP), including male sex, chronic pulmonary diseases, a C-reactive protein level <12.35 mg/dL, and a Pneumonia Severity Index (PSI) score of IV-V [63]. For VAP, *P. aeruginosa* should always be considered [64], with the risk increasing in patients with prior colonization and prolonged hospital stays [65].

Risk factors for MDR-PA infection include prior hospitalization, ICU admission [36,46,66,67,68], local epidemiology with MDR-PA prevalence over 20% [66], prior *P. aeruginosa* colonization [66,69,70], prior exposure to broad-spectrum antibiotics [36,46,66,67,68,70,71,72], the presence of CVC, UC, or tracheostomy [35,47,58,60], diabetes mellitus [66,69], septic shock [69,73], chronic kidney disease [74], and MV [36].

Identifying ICU patients at risk of MDR-PA infection remains challenging due to the overlap of these risk factors [64]. Among these, prior exposure to broad-spectrum antibiotics is the most significant, with the risk increasing with prolonged exposure and the number of antibiotics administered. Notably, for MDR-PA CAP, prior antibiotic treatment is the sole identified risk factor [63], and for VAP, it has the greatest weight [67,68,75]. Lodise et al. highlighted the number of antibiotics received during hospitalization as a predictor of the risk of infection by MDR Gram-negative bacilli, including *P. aeruginosa*, with a significant increase in risk beyond the fourth antibiotic. This study also linked MDR-PA to local *P. aeruginosa* prevalence, prior infection within three months, and infection sites, such as VAP and complicated UTIs [76].

A systematic review identified 38 risk factors for CR-PA infection. These were ranked by experts, with the most critical factors related to prior hospital contact (previous hospitalization or length of stay until infection) and prior antibiotic therapy (e.g., carbapenems, aminoglycosides, cephalosporins, or quinolones). ICU admission, disease severity (measured by SAPS II or APACHE scores), and invasive devices were considered moderately important risk factors [77].

In an effort to quantify risk, a scoring system was proposed to predict CR-PA infection, including variables such as residence in a long-term care facility, tracheostomy presence, prior CR-PA infection within 30 days, hospitalization within the past six months, and treatment with carbapenems, cephalosporins, or quinolones in the last 30 days. The area under the ROC curve (AUC) for this score was 0.81. Another score to identify infections by *P. aeruginosa* resistant to carbapenems, ceftazidime, and piperacillin–tazobactam included similar variables, with an AUC of 0.82 [78].

In general, risk factors and predictive scores show high sensitivity but low specificity, resulting in high negative predictive value but low positive predictive value. Therefore, it is crucial to understand local microbiology and resistance profiles to guide clinical decisions effectively.

## 5. Prognosis

*P. aeruginosa* can sometimes persist in the lungs of patients with chronic pulmonary diseases as an indolent colonizer. However, in other cases, it behaves aggressively, invading tissues and causing severe pneumonia, bacteremia, septic shock, and death. This aggressive behavior is attributed to its ability to produce various toxins and trigger cytokine release pathways associated with sepsis and multiorgan failure [75]. When combined with the challenges of treatment, it becomes evident that the prognosis of *P. aeruginosa* infections is grim.

Mortality from *P. aeruginosa* bacteremia is higher than that caused by other bacteria, even after adjusting for comorbidities and appropriate antibiotic therapy [79], reaching a 30-day mortality rate of 26% [80]. In neutropenic patients, *P. aeruginosa* bacteremia is associated with higher mortality and an increased likelihood of ICU admission [72,81].

Community-acquired pneumonia caused by PES pathogens (*Pseudomonas aeruginosa*, extended spectrum beta-lactamase-producing Enterobacteriaceae, and methicillin-resistant *S. aureus*) is associated with higher 30-day mortality compared to infections caused by other bacteria [82].

MDR-PA infections are linked to faster disease progression, prolonged hospital stays, higher mortality rates [73,83,84,85,86,87], and increased healthcare costs [88]. Compared to non-MDR PA, MDR-PA pneumonia is characterized by longer ICU stays, extended durations of MV, and higher mortality rates, reaching 44% in some studies [6,83,89]. Mortality from *P. aeruginosa* bacteremia ranges between 20% and 39% [79,90,91], with CR-PA bacteremia posing a greater risk of death than non-resistant cases [92]. Although factors related to the virulence of *P. aeruginosa* contribute to the severity of its infections [93], inadequate or second-line treatments are a significant issue. Patients with MDR, XDR, or PDR infections often received suboptimal therapies due to limited treatment options [89,94,95].

## 6. Therapeutical Options for DTR-PA

### 6.1. Aminoglycosides

The rise in the prevalence of nosocomial infections caused by MDR-GNB, particularly MDR, XDR, and DTR-PA, has driven the development of new antibiotics as well as the re-evaluation of “old antibiotics”, including aminoglycosides (AGs) [94].

The most common mechanism of resistance to AGs is the acquisition of transferable genetic elements (plasmids, transposons, and integrons) encoding AG-inactivating enzymes (acetyltransferases, phosphotransferases, and adenyltransferases). Less frequently, resistance arises from 16S rRNA methyltransferases, which confer resistance to all AGs, including plazomicin [96]. Inducible overexpression of MexXY efflux pumps is a less common mechanism of resistance [94]. Tobramycin exhibits the highest intrinsic activity among AGs, but most *P. aeruginosa* strains with enzymes that inactivate gentamicin and tobramycin remain susceptible to amikacin [97,98].

Plazomicin is a new AG recently approved by the FDA for the treatment of pyelonephritis and complicated urinary tract infections (cUTIs) caused by Enterobacteriaceae producing class A, C, and D beta-lactamases. Although plazomicin is not affected by enzymatic inactivation, it does not exhibit greater efficacy against *P. aeruginosa* compared to other AGs [99,100]. In Europe, the pharmaceutical company that had applied for approval for its use in cUTIs withdrew the application in 2020 [101].

#### 6.1.1. In Vitro Activity

In the ASPIRE-ICU study, the analysis of 723 samples from 402 patients across 11 European countries demonstrated *P. aeruginosa* susceptibility rates of 80% for respiratory samples and 90% for perianal samples, according to EUCAST breakpoints [37]. A U.S.-based study observed that 54.3% of MDR-PA isolates were susceptible to gentamicin and 77.6% to tobramycin, according to CLSI breakpoints [102].

Additionally, there is evidence of in vitro synergy between AGs (amikacin and tobramycin) and carbapenems (imipenem or meropenem) [103].

#### 6.1.2. Clinical Experience

In a case series of eleven ICU patients with infections caused by MDR-PA treated with high-dose AGs (seven with amikacin, three with gentamicin, and one with tobramycin, targeting peak concentrations exceeding eight times the MIC) combined with high-flow venovenous hemofiltration (>45 mL/kg/h), a favorable clinical response was achieved at the end of treatment in seven patients (65%) [104].

In recent years, the main use of AGs has been in combination therapies, including with new antibiotics, such as CFT-TAZ or CAZ-AVI [105,106,107]; however, no clinical benefit has been demonstrated, and there is a higher incidence of nephrotoxicity associated with these combinations.

### 6.2. Cefiderocol

Cefiderocol is a siderophore cephalosporin composed of a cephalosporin core (with side chains similar to cefepime and ceftazidime) and a catechol functional group that acts as an iron chelator. This property provides a unique mechanism for penetrating aerobic Gram-negative bacterial cells by chelating extracellular iron and actively transporting it through ferric ion transport systems [108,109,110,111].

The drug’s catechol moiety binds to extracellular trivalent iron and utilizes active transport systems to enter the periplasmic space. Once inside, it dissociates from the iron. Additionally, like other beta-lactam antibiotics, cefiderocol can reach the periplasmic space through simple diffusion via porins [108,112].

Cefiderocol’s structure and cellular entry mechanism allow it to retain activity despite porin channel loss, efflux pump overexpression, and hydrolysis by many beta-lactamases (including MBL) [109,110,111,112,113].

Like other cephalosporins, cefiderocol inhibits cell wall synthesis by binding to penicillin-binding proteins (PBPs), specifically PBP-3.

Cefiderocol is active against MDR GNB, including carbapenem-resistant strains due to carbapenemase production. It is more stable (compared to other beta-lactams, like ceftazidime, cefepime, and meropenem) against class A, B, and D carbapenemases, such as KPC, VIM, IMP, NDM, and OXA, and is effective against *P. aeruginosa* and *Acinetobacter baumannii* [110,111,114].

Cefiderocol shows weak or no activity against Gram-positive bacteria and anaerobes [109].

The FDA has approved cefiderocol for use in cUTIs, including pyelonephritis, nosocomial pneumonia, and VAP caused by susceptible Gram-negative bacilli. The EMA has authorized its use for treating infections due to aerobic Gram-negative microorganisms in adults with limited treatment options.

#### 6.2.1. In Vitro Activity

An in vitro study reported cefiderocol MIC90 values of 1, 2, and 0.5 mg/L for *P. aeruginosa* (including MBL-producing strains), *A. baumannii* (including carbapenem-resistant strains), and *Stenotrophomonas maltophilia*, respectively, compared to meropenem’s MIC90 >16 mg/L [114].

Longshaw et al. analyzed a cohort of carbapenem-resistant Gram-negative bacilli from European patients, reporting *P. aeruginosa* susceptibility to cefiderocol at 98.3%, second only to colistin [115].

In studies of CR-PA isolates (mostly carbapenemase producers), cefiderocol susceptibility rates were 98% (CLSI) and 95% (EUCAST), surpassing CAZ-AVI, CFT-TAZ, IMI-REL, MER-VAB, and amikacin. Activity varied by carbapenemase type: 92–97% (VIM), 87–100% (IMP), 69–85% (NDM), 97–100% (GES), and 100% (KPC) [116].

In an analysis of 7700 *P. aeruginosa* clinical isolates from Europe and North America, cefiderocol susceptibility was 99.9%, 99.8%, 100%, and 99.8% for all isolates, meropenem non-susceptible, CAZ-AVI non-susceptible, and CFT-TAZ non-susceptible strains, respectively. High activity was observed against MBL-producing *P. aeruginosa*, with 97.4% having MICs ≤2 mg/L (EUCAST breakpoint) and 100% ≤4 mg/L (CLSI breakpoint) [117,118]. A separate study reported 100% susceptibility of IMI-REL-resistant *P. aeruginosa* to cefiderocol [119].

#### 6.2.2. Clinical Evidence

In the first RCT, cefiderocol (2 g infused over 1 h every 8 h) demonstrated non-inferiority to imipenem–cilastatin in treating cUTIs caused by Gram-negative bacilli. Among 371 microbiologically evaluable patients, 23 cases involved *P. aeruginosa*, with no differences between treatment groups [120].

The APEKS-NP Trial compared cefiderocol (2 g infused over 3 h every 8 h) to meropenem (2 g infused over 3 h every 8 h) in nosocomial pneumonia, including VAP. Non-inferiority was established, with 48 *P. aeruginosa* cases showing no treatment arm differences [121].

In the CREDIBLE-CR Trial, cefiderocol was compared to the best available therapy (BAT) for carbapenem-resistant Gram-negative infections, including nosocomial pneumonia, VAP, bacteremia, and cUTIs. While clinical and microbiological efficacy was similar to BAT, higher mortality (non-significant) was observed in the cefiderocol group, associated with *A. baumannii* infections. In *P. aeruginosa* infections, mortality was higher with cefiderocol (35% vs. 17%), equalizing after adjusting for *A. baumannii* co-infections (18% vs. 18%) [122].

#### 6.2.3. Clinical Experience

In a compassionate use program for cefiderocol in treating MDR or CR-PA infections, susceptibility was 91%. Clinical response rates were 69% for infections with MIC ≤1 μg/mL, 69% (9/13) for MIC 2–4 μg/mL, and 100% (4/4) for MIC ≥8 μg/mL. All-cause 28-day mortality rates were 23% (MIC ≤1 μg/mL), 33% (MIC 2–4 μg/mL), and 0% (MIC ≥8 μg/mL) [123].

Cefiderocol is well tolerated and has a safety profile similar to other beta-lactams [120,121]. Data support its efficacy and safety in critically ill patients [124].

### 6.3. Ceftazidima–Avibactam

Ceftazidime–avibactam (CAZ-AVI) is an antibiotic combining a third-generation cephalosporin (ceftazidime) with a non-beta-lactam beta-lactamase inhibitor (avibactam). Ceftazidime exhibits bactericidal activity by inhibiting bacterial peptidoglycan cell wall synthesis through binding to penicillin-binding proteins (PBPs), leading to cell death. Avibactam inhibits the activity of Ambler class A (e.g., ESBLs and KPC), class C (e.g., AmpC), and some class D (e.g., OXA-48) beta-lactamases but does not inhibit class B enzymes (e.g., MBLs) or other class D enzymes (e.g., OXA-23, OXA-24). Avibactam does not protect against other resistance mechanisms, such as porin mutations (e.g., OprD) or efflux pump overexpression. Consequently, CAZ-AVI is ineffective in the presence of these resistance mechanisms [1,125,126].

With this profile, CAZ-AVI is active against most carbapenem-resistant *Enterobacteriaceae* and MDR-PA, making it a potential alternative to carbapenems for severe infections caused by Gram-negative MDR bacteria [127,128].

CAZ-AVI is approved by the FDA and the EMA for treating hospital-acquired bacterial pneumonia (including VAP), cUTIs (including pyelonephritis), and complicated intra-abdominal infections (cIAIs) in combination with metronidazole. EMA also approved CAZ-AVI for infections caused by aerobic Gram-negative bacteria with limited treatment options.

#### 6.3.1. In Vitro Activity

Studies report that 95.5–98% of *P. aeruginosa* isolates are sensitive to CAZ-AVI. For CR-PA, sensitivity ranges between 43 and 83.3%, depending on resistance mechanisms.

Candel et al. analyzed over 20,000 Gram-negative isolates from 24 European countries (2013–2018) and found 98% sensitivity of *P. aeruginosa* to CAZ-AVI across different infection sites: bloodstream infections (97.9%), nosocomial pneumonia (98%), cIAIs (99.1%), and cUTIs (98.7%). CR-PA showed 43.2% sensitivity to CAZ-AVI [54].

Sader et al. evaluated the in vitro activity of new beta-lactams (including CAZ-AVI) against *P. aeruginosa* recovered from patients with pneumonia and SSTIs, finding sensitivity rates of 95.5% for pneumonia and 98.3%, respectively. Avibactam restored sensitivity in 79.8–87.5% of ceftazidime-resistant *P. aeruginosa*. Sensitivity varied by carbapenem and geographic region, with rates of 78–92.6% for imipenem-resistant and 56–81.6% for meropenem-resistant *P. aeruginosa* [49,129].

#### 6.3.2. Clinical Evidence

Clinical trials evaluating CAZ-AVI included limited data on infections caused by MDR/XDR *P. aeruginosa* [130,131,132,133,134].

The RECAPTURE Trial compared CAZ-AVI (2.5 g every 8 h) with doripenem (500 mg every 8 h) in cUTIs, including pyelonephritis. Among 810 patients with microbiological results, only 38 had *P. aeruginosa* infections. Efficacy and safety were comparable between treatments [130].

The REPRISE Trial assessed CAZ-AVI (2.5 g every 8 h, plus metronidazole for abdominal infections) versus the best available therapy (BAT, 97% carbapenems) in 333 patients with cIAIs or cUTIs caused by ceftazidime-resistant Enterobacteriaceae or *P. aeruginosa*. CAZ-AVI was a viable carbapenem alternative without differences in efficacy or safety. Only 21 *P. aeruginosa* infections were included [132].

The RECLAIM Trial compared CAZ-AVI (2.5 g every 8 h, with metronidazole) to meropenem (1 g every 8 h) in 1066 patients with cIAIs, including 66 with *P. aeruginosa* infections. Both treatment arms showed similar efficacy and safety, including against ceftazidime-resistant isolates [133,134].

The REPROVE Trial, a phase 3 study in hospital-acquired pneumonia, including VAP, enrolled 879 patients randomized to CAZ-AVI (2.5 g every 8 h) or meropenem (1 g every 8 h). CAZ-AVI was non-inferior in efficacy and safety. Of the Gram-negative pathogens identified, 33.9% were *P. aeruginosa*, but carbapenem-resistant isolates were excluded [131].

Post hoc analyses confirmed CAZ-AVI’s efficacy and safety in treating MDR infections caused by *P. aeruginosa* and *Enterobacteriaceae* [135,136].

#### 6.3.3. Clinical Experience

Although high-quality evidence for CAZ-AVI in treating DTR-PA infections is limited, smaller studies suggest clinical efficacy.

Corbella et al. reviewed 61 MDR/XDR PA infections treated with CAZ-AVI. Common infection sites included pneumonia [34.4%], with 50.8% of patients meeting sepsis criteria. Clinical cure at 14 days was 54.1%, recurrence at 90 days was 12.5%, and 30-day mortality was 13.1%. Monotherapy was associated with better outcomes (14-day cure rate 63.5%, recurrence 6.5%, and mortality 6.3%) without detected resistance development [137].

Xu et al. analyzed 84 cases of severe hospital-acquired pneumonia in ventilated patients with DTR-PA (53.6%) and CR-PA isolates [46.4%]. Clinical cure was 63.1%, and 30-day mortality was 18.9%. Outcomes improved with loading doses and prolonged infusion regimens but not with combination treatment [138].

Almangour et al. compared CAZ-AVI to CFT-TAZ in 200 patients with MDR-PA, including critically ill patients (56%). Clinical cure and 30-day mortality rates were comparable between CAZ-AVI (66% cure, 23% mortality) and CFT-TAZ (61% cure, 27% mortality) [139].

### 6.4. Ceftolozane–Tazobactam

Ceftolozane–tazobactam (CFT-TAZ) is an antibiotic that combines a fourth-generation cephalosporin (ceftolozane) with a beta-lactamase inhibitor (tazobactam). Ceftolozane differs from other cephalosporins due to its potent activity against *P. aeruginosa*, achieved through modifications in its side chains at positions 3 and 7 of the dihydrothiazine ring. These modifications confer increased stability against AmpC-type chromosomal beta-lactamase-producing *P. aeruginosa* and protect it from the effects of OprD porin loss or efflux pump mutations. Tazobactam extends activity against most ESBL-producing organisms and some anaerobes (e.g., *Bacteroides* spp.) [140]. However, it lacks activity against class A beta-lactamases, such as KPC, as well as class B and class D beta-lactamases. It is also inactive against *Stenotrophomonas* spp. and *Acinetobacter* spp [141].

CFT-TAZ exhibits high activity against *P. aeruginosa*, including XDR strains, and retains activity against ESBL-producing Enterobacteriaceae and certain *Streptococcus* species. It has been approved by the FDA and EMA for the treatment of nosocomial pneumonia, including VAP, cUTIs, including pyelonephritis, and complicated intra-abdominal infection cIAIs in combination with metronidazole.

#### 6.4.1. In Vitro Activity

CFT-TAZ demonstrates in vitro activity against 93.5–97.5% of *P. aeruginosa* isolates, with activity against carbapenem-resistant *P. aeruginosa* ranging from 44.1% to 75.4%, depending on the study. Candel et al. reported a *P. aeruginosa* sensitivity rate of 97.5% for CFT-TAZ. Sensitivity rates for different clinical sources included 96.7% in bacteremia, 97.5% in NN, 99.1% in cIAIs, and 97.6% in cUTIs. Among carbapenem-resistant *P. aeruginosa*, activity decreased to 44.1% [54].

Sader et al. reported a sensitivity rate of 93.3% for *P. aeruginosa* strains isolated from pneumonia. CFT-TAZ retained activity against 81.9% of imipenem-resistant strains and 57.1% of meropenem-resistant strains [49].

Shortridge et al. evaluated *P. aeruginosa* susceptibility to CFT-TAZ from various clinical sources globally (America, Europe, and Asia), including MDR and XDR isolates. They reported an overall sensitivity rate of 93.5%, without significant geographic differences. For MDR and XDR *P. aeruginosa* strains, sensitivity rates were 69.2% and 59.1%, respectively, with meropenem-resistant strains showing a sensitivity of 75.4% [142].

#### 6.4.2. Clinical Evidence

In the ASPECT-cUTI Trial, CFT-TAZ was compared to levofloxacin for the treatment of cUTIs, showing superior outcomes; however, *P. aeruginosa* was detected in only 19 of 1083 patients included in the study [143].

The ASPECT-cIAI Trial included 993 patients with cIAIs, of whom 75 had *P. aeruginosa* infections. CFT-TAZ (1.5 g every 8 h) combined with metronidazole (500 mg every 8 h) was compared to meropenem (1 g every 8 h), demonstrating non-inferiority. *P. aeruginosa* isolates showed sensitivity rates of 98.6% to CFT-TAZ and 89.9% to meropenem [144].

The ASPECT-NP Trial involved 726 patients with nosocomial pneumonia (71% VAP), with *P. aeruginosa* isolated in 128 cases (39% MDR or XDR). Patients were randomized to receive CFT-TAZ (3 g every 8 h) or meropenem (1 g every 8 h). The study demonstrated the non-inferiority of CFT-TAZ compared to meropenem for nosocomial pneumonia treatment [145]. A subgroup analysis of ventilated nosocomial pneumonia patients (n = 207) showed 28-day mortality rates of 24.2% for CFT-TAZ versus 37% for meropenem (not significant); specifically, for *P. aeruginosa*-related nosocomial pneumonia, mortality rates were 11.8% and 29.4%, respectively [146].

#### 6.4.3. Clinical Experience

Balandín et al. conducted a multicenter observational study of 95 critically ill patients treated with CFT-TAZ for severe *P. aeruginosa* infections, primarily nosocomial pneumonia (56.2%). Among these patients, 49.5% had sepsis, 45.3% had septic shock, and 36.8% had MDR-PA. Despite the severity, clinical response was favorable in 71.6% of cases, with a 30-day mortality rate of 30.5%. Monotherapy was used in 44.2% of cases, with no observed benefits of combination therapy [147].

Pogue et al. compared CFT-TAZ with polymyxins or aminoglycosides for MDR or XDR-PA infections in critically ill patients (69% ICU, 63% on MV, and 42% with septic shock). CFT-TAZ showed higher cure rates and reduced renal failure without affecting overall mortality [105].

Almangour et al. reported higher cure rates and improved safety (lower renal failure rates) with CFT-TAZ compared to colistin for MDR-PA infections, although the difference in mortality (39% vs. 49%) was not statistically significant [148].

The cohort study comparing CFT-TAZ with CAZ-AVI has been previously discussed [139].

CFT-TAZ has accumulated substantial clinical efficacy evidence, with several case series reporting favorable outcomes (cure rates between 63% and 83%) in MDR or CR-PA infections [106,149,150,151,152].

### 6.5. Colistin

Polymyxins are a class of antimicrobials that include five chemically distinct compounds (polymyxins A, B, C, D, and E), of which polymyxin B and colistin (polymyxin E) are the only ones currently available on the market [153,154,155].

Colistin was approved by the FDA in 1959 for the treatment of infections caused by Gram-negative bacteria, but its high toxicity limited its use. It is a highly nephrotoxic agent, and acute kidney injury frequently occurs at standard doses [156]. Additionally, colistin is associated with neurotoxic effects, such as dizziness, muscle weakness, facial and peripheral paresthesia, vertigo, confusion, ataxia, and neuromuscular blockade, which could lead to respiratory failure or apnea, seizures, and even coma [157]. The emergence of MDR-GNB bacteria renewed interest in colistin during the 1990s.

Colistin disrupts the bacterial membrane due to its positive charge, causing electrostatic attraction to the anionic components of magnesium and calcium that stabilize the membrane. This interaction results in the leakage of cellular contents and, ultimately, cell death Moreover, high doses of colistin reduce endotoxin and cytokine levels in sepsis [158].

The antimicrobial spectrum of colistin includes MDR and XDR GNB, regardless of the resistance mechanism, primarily *Klebsiella pneumoniae*, *Acinetobacter baumannii*, and *Pseudomonas aeruginosa*. However, *Stenotrophomonas maltophilia*, *Proteus* spp., *Providencia* spp., *Serratia* spp., and *Burkholderia* are intrinsically resistant.

Colistin is active against metabolically inactive cells located in the inner layers of biofilms, which may complement the activity of antimicrobials effective only against metabolically active cells in the biofilm. This activity has been confirmed in cystic fibrosis patients using nebulized therapy [159,160].

#### 6.5.1. In Vitro Activity

Despite decades of use, colistin retains high in vitro activity against Enterobacteriaceae and non-fermenting GNB, even in MDR bacteria. Longshaw et al. reported a sensitivity rate of 100% for *P. aeruginosa* (MDR and CR-PA), the highest among the antibiotics analyzed in their study [115]. Satlin et al. examined 46 infections caused by MDR and CR-PA and found only one resistant strain to colistin [123]. Delgado-Valverde et al. observed similar results, with 86% of 399 *P. aeruginosa* isolates resistant to imipenem being susceptible to colistin, with an MIC90 of 4 mg/L [161]. Carvalhaes et al. reported a 99.1% susceptibility of MDR-PA to colistin, with an MIC90 of 2 mg/L [162].

#### 6.5.2. Clinical Evidence

A meta-analysis published in 2015 evaluated the treatment of respiratory infections caused by *P. aeruginosa* and *A. baumannii*. This analysis, which included nine studies (none of which were RCTs) with significant heterogeneity, suggested that the efficacy and side effects (particularly nephrotoxicity) were comparable to other available alternatives at that time [163].

However, more recent and higher-quality studies challenge this assertion. The Magic Bullet RCT compared the empirical treatment of VAP with colistin (loading dose of 4.5 million IU followed by 3 million IU every 8 h) or meropenem (2 g every 8 h), and both were combined with levofloxacin. It demonstrated that colistin was less effective than meropenem and carried a higher risk of nephrotoxicity [164].

The combination of colistin with meropenem in the treatment of infections caused by MDR GNB does not improve outcomes, reduce side effects, or prevent resistance development [165,166,167]. However, clinical guidelines recommend its use in combination with one or more additional antibiotics against CR-PA due to doubts about its standalone efficacy [168].

The RESTORE-IMI 1 Trial compared the combination of colistin and imipenem with imipenem–relebactam in an RCT involving infections caused by imipenem-resistant bacteria (predominantly *P. aeruginosa*). The results favored imipenem–relebactam in terms of clinical response, mortality, and nephrotoxicity [169].

#### 6.5.3. Clinical Experience

Several case series have documented outcomes in patients with infections caused by MDR GNB treated with colistin, including *P. aeruginosa*, with cure rates of 74.6–79.1% and nephrotoxicity rates of 7.6–10% [170,171]. These results are acceptable, particularly in the absence of other therapeutic options. However, newer antibiotics with activity against MDR/XDR/DTR *P. aeruginosa* and improved safety profiles have been developed in recent years.

In a cohort study by Pogue et al., outcomes of treatments based on CFT-TAZ were compared with those based on colistin or aminoglycosides for MDR-PA infections (69% of patients in ICUs, 52% with VAP, and 42% meeting sepsis/shock criteria). The results were unfavorable for patients treated with colistin or aminoglycosides, showing lower clinical cure rates, higher risks of renal failure, and greater mortality [105]. Almangour et al. reported similar findings [148].

### 6.6. Fosfomycin

Fosfomycin, first marketed in 1969, is another antibiotic revived due to the emergence of MDR GNB [172].

It is a bactericidal antibiotic that inhibits the enzyme pyruvyltransferase (MurA), which is essential for the first step of peptidoglycan synthesis, making cross-resistance with other antibiotics unlikely [173]. Fosfomycin achieves high concentrations in serum and urine and penetrates well into various body compartments. Its activity may increase in acidic and anaerobic conditions, and it exhibits a post-antibiotic effect. The primary risks associated with its use are hypokalemia and sodium overload [172]. It has been used in the treatment of MDR or DTR *P. aeruginosa* infections as part of combination therapy regimens [94].

Resistance mechanisms to fosfomycin include reduced drug uptake due to mutations in transporters or alterations in biological systems regulating their expression, modifications of MurA, or the presence of fosfomycin-inactivating enzymes encoded by the fosA and fosB genes [174].

#### 6.6.1. In Vitro Activity

The SENTRY surveillance program analyzed 141 *P. aeruginosa* isolates and found that 97.2% had an MIC ≤256 mg/L [175]. Additionally, 86.4% of *P. aeruginosa* strains evaluated in a Spanish multicenter study were susceptible, with an ECOFF of 128 mg/L [176].

A systematic review examining the role of fosfomycin in treating MDR non-fermenting GNB found that 30.2% of the 1693 MDR-PA isolates analyzed were susceptible [177]. Synergistic effects have been observed with beta-lactams, aminoglycosides, or quinolones in 53.5% of MDR-PA isolates [177], and synergy has also been demonstrated with ceftazidime–avibactam CAZ-AVI [178] and meropenem, with 100% of MDR-PA isolates showing synergy according to Albiero et al. [179].

Resistance emergence during treatment is the most feared complication and one of the reasons for recommending its use as part of a combination therapy [172].

#### 6.6.2. Clinical Experience

A systematic review by Falagas et al. included only 33 patients with MDR-PA treated with fosfomycin (25 of them in combination therapy), reporting favorable outcomes in 90.9% of cases [177].

A study conducted in Thailand involving patients with pneumonia caused by CR-PA with intermediate sensitivity to doripenem compared two treatment regimens: high-dose doripenem in a 4 h infusion combined with fosfomycin versus colistin combined with fosfomycin in 49 patients. No differences were observed in mortality, clinical or microbiological cure, or adverse effects between the two groups [180].

Additionally, a case of meningitis caused by XDR-PA was successfully treated with high-dose CFT-TAZ and fosfomycin [181].

### 6.7. Imipenem–Relebactam

The most common resistance mechanisms to imipenem include the production of carbapenemases and the combination of reduced permeability due to porin alterations (OprD) with the overexpression of AmpC or ESBLs.

Relebactam is a potent non-beta-lactam beta-lactamase inhibitor. Its structure is related to that of avibactam but is distinguished by the addition of a piperidine ring to the carbonyl group at position 2. Relebactam’s high reactivity, resulting from its chemical structure, leads to limited stability in the presence of bases or nucleophiles but simultaneously makes it a potent beta-lactamase inhibitor [182].

This novel combination of a beta-lactam and a beta-lactamase inhibitor has a broad spectrum of activity against Ambler class A beta-lactamases, including ESBLs and KPCs, as well as class C beta-lactamases (AmpC). However, it is ineffective against class B and class D carbapenemases [183,184].

The EMA has approved IMI-REL for hospital-acquired respiratory infections (nosocomial pneumonia and VAP), including secondary bacteremia, and infections caused by GNB when other treatments may be ineffective. The FDA has approved it for cUTI and cIAI.

#### 6.7.1. In Vitro Activity

Adding relebactam reduces the MIC90 of imipenem-resistant *P. aeruginosa* isolates from 32 mg/L to 8 mg/L [161], restoring imipenem activity against 80% of imipenem non-susceptible *P. aeruginosa* strains [185].

IMI-REL has shown in vitro activity against 93.7% of 474 *P. aeruginosa* isolates from ICU patients in Spain and Portugal, including 94.1% of MDR-PA, 78.8% of XDR-PA, and 75% of DTR-PA [186].

It has also demonstrated high activity against *P. aeruginosa* strains resistant to CAZ-AVI or CFT-TAZ [187].

IMI-REL has demonstrated up to five times higher activity than imipenem alone against non-carbapenemase-producing CR-PA thanks to its high activity against the presence of AmpC-type beta-lactamases associated with impermeability and also against KPC-producing strain [188,189]. By adding relebactam, the proportion of CR-PA isolates (non-carbapenemase producers) susceptible to imipenem increases from 2% to 63%. Among these isolates, 98% and 80% are also susceptible to colistin and CFT-TAZ, respectively [190].

A Spanish study identified IMI-REL as the antibiotic with the highest activity against *P. aeruginosa*, with 97.3% effectiveness [191].

#### 6.7.2. Clinical Evidence

In the RESTORE-IMI 1 Trial, IMI-REL was superior to the combination of colistin and imipenem for treating infections caused by imipenem-resistant bacteria (predominantly *P. aeruginosa*) in terms of clinical response, mortality, and nephrotoxicity [169].

The RESTORE-IMI 2 Trial demonstrated the non-inferiority of IMI-REL compared to piperacillin–tazobactam in treating nosocomial pneumonia, including VAP. Among 537 pneumonia cases included, 19% were caused by *P. aeruginosa*. In this subgroup, the all-cause mortality of patients treated with IMI-REL was 33.3%, with a favorable clinical response rate of 46.7%, compared to 12.0% and 68.0%, respectively, for infections treated with piperacillin–tazobactam [192].

#### 6.7.3. Clinical Experience

There are limited real-world studies on the efficacy of IMI-REL. Recently, Leanza et al. published a single-center case series including five patients with DTR-PA infections treated with IMI-REL, reporting favorable outcomes [193].

### 6.8. Meropenem–Vaborbactam

Vaborbactam is an innovative beta-lactamase inhibitor characterized by its boron-ring structure, which provides significant activity against serine beta-lactamases, particularly KPC enzymes. This structural feature has been specifically developed to enhance the activity of carbapenems [194]. While vaborbactam lacks intrinsic antibacterial activity, its combination with meropenem significantly improves efficacy against GNB, particularly *Enterobacteriaceae* producing class A and C carbapenemases. However, this combination is ineffective against OXA-48 beta-lactamases and MBL [195].

MER-VAB has been approved by the FDA for the treatment of cUTIs and by the EMA for the treatment of cUTIs, cIAIs, and nosocomial pneumonia, including VAP. It is also indicated for infections caused by GNB when other antibiotics may not be appropriate.

#### 6.8.1. In Vitro Activity

The contribution of vaborbactam to meropenem’s activity against *Pseudomonas aeruginosa* is limited, as it does not counteract reduced permeability or increased efflux pump activity associated with meropenem resistance.

However, the overexpression of MexXY efflux pumps and AmpC enzymes may result in lower MICs for MER-VAB compared to meropenem alone [196]. In a study by Carvalhaes et al., MER-VAB was active against 59% of MDR-PA isolates recovered from pneumonia cases in U.S. hospitals, compared to 22.1% susceptibility to meropenem alone. This highlights the importance of understanding the microbiology and resistance mechanisms prevalent in specific units [162].

MER-VAB demonstrated activity against 82.1% of *P. aeruginosa* isolates (vs. 67.3% for meropenem alone) and 41.0% of MDR isolates (vs. 13.0% for meropenem alone) recovered from pneumonia (including VAP) cases in Europe [197].

#### 6.8.2. Clinical Evidence

RCTs conducted with meropenem have not included infections caused by *P. aeruginosa* [198,199].

#### 6.8.3. Clinical Experience

Published clinical experience with MER-VAB for *P. aeruginosa* infections is limited.

In a real-world analysis by Alosaimy et al., MER-VAB was used to treat eight MDR-PA infections, reporting a 30-day mortality rate of 0% and a 90-day mortality rate of 12.5% [200].

### 6.9. New Antibiotics

In addition to those previously described, other antibiotics have been studied and will soon be available to us [141].

#### 6.9.1. Aztreonam–Avibactam

Aztreonam is stable against class B carbapenemases, and avibactam inhibits the action of class A, C, and D beta-lactamases. Aztreonam–avibactam is approved by the EMA for the treatment of cIAI and cUTI, nosocomial pneumonia, and infections caused by aerobic Gram-negative bacteria with limited treatment options.

The in vitro activity against *P. aeruginosa* is lower compared to other antibiotics, like CAZ-AVI, CFT-TAZ, or IMI-REL [201], but it may be useful for *P. aeruginosa* producing MBLs, like VIM or IMP [202,203].

The REVISIT Trial compared the treatment of patients with cIAI or nosocomial pneumonia/VAP with meropenem (with or without colistin) to treatment with aztreonam–avibactam. The results of both treatments were comparable, but patients with monomicrobial *P. aeruginosa* infections were excluded from this study [204].

With the data available now, treatment with aztreonam–avibactam seems a more favorable option for treating Enterobacteriaceae than for PA [205].

#### 6.9.2. Cefepime–Enmetazobactam

Cefepime–Enmetazobactam has shown good results in an RCT when compared to piperacillin–tazobactam for the treatment of cUTIs [206]. However, the contribution of Enmetazobactam against the resistance mechanisms of *P. aeruginosa* is limited and does not enhance the spectrum of cefepime [207,208].

#### 6.9.3. Cefepime–Taniborbactam

Taniborbactam is a potent beta-lactamase inhibitor with activity against class A, B, C, and D beta-lactamases. It has been shown to achieve 95% susceptibility against MDR-PA recovered from bacteremias in cancer patients, which is higher than its comparators (70% ceftazidime–avibactam, 65% ceftolozane–tazobactam, 65% amikacin) [209].

Cefepime–taniborbactam is active in vitro against 85.3% of meropenem-resistant *P. aeruginosa*, 79.6% of MER-VAB-resistant *P. aeruginosa*, 71.3% of CAZ-AVI-resistant *P. aeruginosa*, and 70.1% of CFT-TAZ-resistant *P. aeruginosa* [210]. It has been shown to be effective against 60% of carbapenemase-producing *P. aeruginosa* and 55.6% of MBL-producing strains, with 61.2–81.9% susceptibility to VIM and 93.3–98.8% to GES [210,211]. *P. aeruginosa’s* resistance to cefepime–taniborbactam arises due to the accumulation of resistance mechanisms, such as porin mutations and PBP alterations, alongside carbapenemase production [210,212].

The CERTAIN-1 Trial showed better results (in a composite outcome of clinical and microbiological success) for patients with cUTIs treated with cefepime–taniborbactam compared to those treated with meropenem [213]. In this RCT, only twenty-three infections caused by *P. aeruginosa* were detected (sixteen in the cefepime–taniborbactam group, seven in the meropenem group), with cure rates above 80% [214]. The only patient with a *P. aeruginosa* infection caused by VIM, treated with cefepime–taniborbactam achieved clinical cure and microbiological eradication.

#### 6.9.4. Cefepime–Zidebactam

The combination of cefepime with zidebactam, a beta-lactam enhancer, has shown efficacy in vitro against Enterobacteriaceae and *P. aeruginosa*. This combination maintains its activity in vitro in the presence of efflux pumps or the downregulation of OprD, but its MIC increases in the presence of AmpC [215]. Cefepime–zidebactam is active in vitro against a high percentage (78–97%) of CR-PA [215,216], including MBL producers [217], and against *P. aeruginosa* resistant to CAZ-AVI or CFT-TAZ [187].

Currently, there is limited clinical data on the activity of cefepime–zidebactam, mainly based on published case reports, where it is described as an effective rescue treatment for *P. aeruginosa* producing NDM [218,219,220].

#### 6.9.5. Meropenem–Nacubactam

The efficacy of combining nacubactam, a non-beta-lactam beta-lactamase inhibitor, with meropenem is under investigation. Nacubactam is effective at inhibiting class A and C beta-lactamases and, to a lesser extent, class D. No clinical data are available yet, but the results from animal models are promising [221,222].

Figure 1 summarizes the activity of each antibiotic against each *P. aeruginosa* resistance mechanism.

## 7. Strategies to Optimize Antibiotic Treatment

The capacity of *P. aeruginosa* to develop antimicrobial resistance necessitates optimizing therapeutic regimens [223]. A critical factor is the accurate dosing of antibiotics. In Table 2, we can see the dosage of the reviewed antibiotics.

Additionally, various strategies have been evaluated to optimize the performance of these antibiotics, including prolonged infusion, combination therapy, and inhaled administration.

### 7.1. Prolonged Infusions

Prolonged, extended, and continuous infusions are strategies designed to achieve more aggressive pharmacokinetic/pharmacodynamic (pK/pD) targets for time-dependent antibiotics, such as beta-lactams. The pK/pD index that best predicts therapeutic success for these antibiotics is the percentage of time during which drug concentrations remain above the MIC (%*f*T > MIC). Traditionally, the target was set at 40–70% of the dosing interval (40–70% *f*T > MIC). However, maintaining drug concentrations at least four times above the MIC for the entirety (or most) of the dosing interval (100% T > 4 × MIC) has been shown to enhance efficacy. Some studies suggest that trough concentrations 3.5–5 times above the MIC could minimize resistance development in infections caused by Gram-negative bacteria [224,225,226].

Although definitive evidence linking extended infusions to reduced mortality is lacking, improved outcomes have been observed in critically ill patients and severe infections (e.g., respiratory infections or SOFA ≥ 9) [227], findings supported by several meta-analyses [228,229,230]. However, the MERCY Trial, published in 2023, included over 600 critically ill patients (20% with *P. aeruginosa* infections) and found no significant differences in mortality or resistance emergence between continuous and intermittent meropenem administration [231]. Prolonged infusion regimens are safe, and even reduced neurotoxicity has been reported with continuous cefepime infusion [232].

Extended infusions (3 h) of CFT-TAZ, CAZ-AVI, and cefiderocol are routinely recommended by clinical guidelines for treating severe infections with high bacterial burdens [45,233,234,235]. For infections caused by *P. aeruginosa* with elevated MICs or in sites with suboptimal drug penetration, prolonged infusion facilitates achieving pK/pD targets [236,237].

Although clinical superiority has not been conclusively demonstrated for prolonged CFT-TAZ infusions, theoretical advantages exist. Pilmis et al. prospectively analyzed 72 *P. aeruginosa*-infected patients (79% ICU, 66.7% respiratory focus), comparing intermittent (<1 h), extended (4 h), and continuous infusions of CFT-TAZ (2/1 g every 8 h). They found that intermittent infusions (<1 h) failed to achieve the 100% ƒT > 4 × MIC target when MICs were ≥4 mg/L, while continuous infusion reached the target even at MICs ≥ 8 mg/L [238].

Similarly, Montero et al. investigated in vitro the behavior of CFT-TAZ against three XDR-PA strains (MICs: 2, 8, 16 mg/L) under intermittent, extended, and continuous infusions. Continuous infusion provided the greatest reduction in bacterial density, especially in strains with higher MICs [239].

For CAZ-AVI, limited data are available regarding prolonged infusion’s impact on *P. aeruginosa* infections. However, Tumbarello et al. found that continuous infusion of CAZ-AVI reduced mortality in treating *Klebsiella pneumoniae* KPC producers [240]. Continuous infusion of CAZ-AVI also increases the likelihood of achieving pK/pD targets with lower doses in MDR Gram-negative infections [241].

A systematic review suggests the optimal dosing of fosfomycin as an 8 g loading dose followed by 16–24 g of continuous infusion every 24 h [242].

Despite limited evidence, a 2023 consensus document recommends extended or continuous beta-lactam infusions with a loading dose in critically ill patients, particularly for Gram-negative infections, to improve cure rates and reduce mortality. The document also suggests individualized therapeutic drug monitoring (TDM) for beta-lactams, although routine TDM is not universally recommended [243].

### 7.2. Combination Therapy

Combining antibiotics aims to achieve synergistic effects, improving patient outcomes or reducing the risk of resistance selection. Although evidence is limited, most studies report additive or synergistic effects when combining CFT-TAZ, CAZ-AVI, IMI-REL, or cefiderocol with agents like amikacin, colistin, fosfomycin, meropenem, or aztreonam [244,245,246].

CFT-TAZ shows synergy with aminoglycosides, even against CFT-TAZ-resistant and GES carbapenemase-producing strains [244,247,248]. An in vitro study using CFT-TAZ and meropenem against XDR-PA (ST 175) strains maintained bacterial suppression over 14 days, whereas monotherapy failed [249].

In in vitro mortality curve studies, a synergistic or additive effect of CAZ-AVI with colistin, aztreonam, amikacin, fosfomycin, and meropenem was observed, even in strains resistant to CAZ-AVI [250,251]. In the study by Mikhail et al., the combination of CAZ-AVI with meropenem demonstrated the greatest reduction in the MICs of CAZ-AVI against 21 MDR-PA isolates [250]. The combination of CAZ-AVI with imipenem or fosfomycin showed similar results against XDR-PA and MDR-PA (non-MBL producers) [178,252]. Against 18 *P. aeruginosa* strains producing GES-type carbapenemases, the combination of CAZ-AVI and meropenem has proven to be an alternative treatment, demonstrating activity against 100% of the strains [253].

The combination of CAZ-AVI and aztreonam showed in vitro efficacy against MBL-producing *P. aeruginosa* strains [251,254]. However, clinical evidence is very limited, and the data published in a systematic review conclude that the efficacy of this combination is greater against Enterobacteriaceae than against *P. aeruginosa* [205].

Synergy has been observed in the combination of CAZ-AVI with aztreonam and amikacin, and there are published cases in which its clinical efficacy against VIM-producing *P. aeruginosa* has been confirmed [255].

Gatti et al. found that continuous fosfomycin combined with extended cefiderocol or continuous CAZ-AVI achieved optimal pK/pD targets and microbiological eradication in five out of six patients with DTR-PA infections [256]. There is evidence to suggest that fosfomycin is more effective when used as part of a combination regimen [257].

A 2021 meta-analysis associated CFT-TAZ combination therapy with reduced all-cause mortality in Gram-negative infections but could not establish a clear link between combination therapy and improved clinical outcomes [258]. Another review comparing CAZ-AVI monotherapy and combination therapy in carbapenem-resistant Enterobacterales and *P. aeruginosa* found no differences in mortality, clinical cure, or resistance development [259].

To sum up, the use of prolonged infusions and/or the combination of synergistic drugs may be helpful in managing DTR-PA infections with low sensitivity to new beta-lactams and high inoculum. However, the available evidence is limited and of low quality.

### 7.3. Inhaled/Nebulized Therapy

Despite theoretical benefits, such as achieving high pulmonary antibiotic concentrations [260], the clinical advantages of inhaled antibiotic therapies remain unproven, and treatment guidelines do not recommend their use [45,235]. Most studies predate the introduction of newer beta-lactams and included heterogeneous populations with *P. aeruginosa* strains often only susceptible to colistin and aminoglycosides [261].

A 2011 systematic review limited the potential benefits of aerosolized therapy to inhaled colistin or aminoglycosides (always combined with systemic antibiotics) in treating MDR bacteria or poorly responding pneumonia [262], and findings were reiterated in 2017 [263].

Nebulized amikacin combined with systemic therapies in VAP was evaluated in a 2021 meta-analysis, showing improved microbiological eradication but no benefits in mortality, MV duration, or ICU length of stay [264]. The IASIS trial found no clinical benefits despite reduced bacterial loads with inhaled amikacin and fosfomycin [265].

The lack of benefit may stem from formulations not optimized for inhalation, compromising lung tissue penetration, even with appropriate nebulization devices. Furthermore, inhaled/nebulized therapies are not without adverse effects, primarily bronchospasm.

## 8. Guidelines and Recommendations

IDSA guidelines recommend the treatment of non-urinary tract infections caused by DTR-PA with ceftolozane–tazobactam, ceftazidime–avibactam, or imipenem–relebactam as the first options. They suggest cefiderocol as an alternative, which is the preferred option in cases of MBL production. If DTR-PA is susceptible to any of these antibiotics, monotherapy is recommended [266].

ESCMID guidelines, endorsed by ESICM, do not establish recommendations on the preferred treatment or new antibiotics and recommend using combination therapy in the case of severe CR-PA infections if treatment with aminoglycosides, colistin, or fosfomycin is required [235].

## 9. Treatment Proposal

When prescribing empirical antimicrobial treatment for an infection that may be caused by a DTR-PA in a critically ill patient, it is important to consider the microbiology of our ICU and the patient’s colonization status.

As we have seen, risk factors and scores have low predictive value, so, before the microbiological results are available, it is not possible to determine whether the infection (for example, a VAP) is caused by *P. aeruginosa* or another GNB. Therefore, the resistance mechanisms present in the GNB in our unit must be taken into account. When establishing a threshold of 10%, as recommended by other authors [267], if the prevalence of carbapenemase-producing GNB exceeds 10% in our setting, we should prescribe treatment active against these carbapenemases. If it is below this 10%, our first option would be CFT-TAZ.

Similarly, if a critically ill patient is colonized by an MDR GNB and develops a severe infection, this must be considered, and treatment should be active against the bacteria colonizing the patient.

Figure 2 shows the proposed empirical treatment algorithm.

Once the microbiological results are available, our order of preference, considering efficacy, the impact on antimicrobial flora, and antibiotic stewardship, would be CFT-TAZ, IMI-REL, CAZ-AVI, and cefiderocol (as long as its in vitro sensitivity has been demonstrated and regardless of the selected empirical treatment).

Both empirical and targeted treatments can be given as monotherapy, provided it is possible to treat with one of the first-line antibiotics (CFT-TAZ, IMI-REL, CAZ-AVI, or cefiderocol).

## 10. Conclusions

Infection by DTR-PA is a challenge in critically ill patients due to its severity or the difficulty in selecting an appropriate antibiotic treatment, both empirically and even when targeted. In this review, we have analyzed the available therapeutic options and propose a therapeutic algorithm to facilitate decision making, which should always be based on local resistance patterns.

## Figures and Tables

**Figure 1 antibiotics-14-00178-f001:**
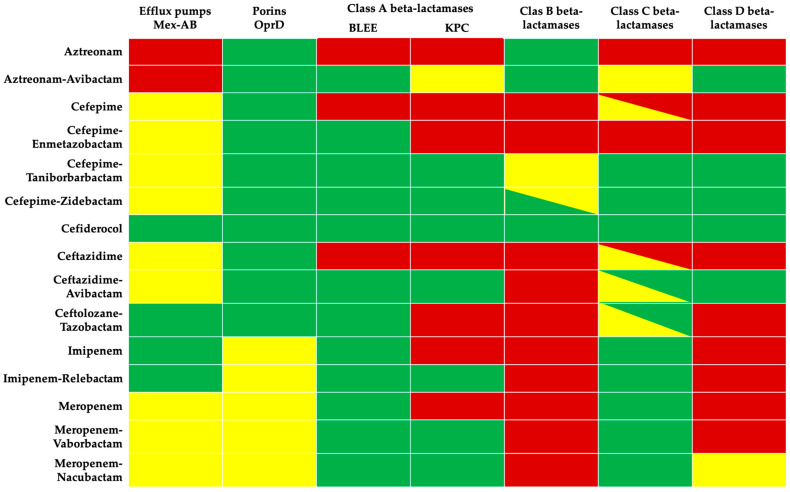
Antibiotic activity against *P. aeruginosa* resistance mechanisms (green: high activity, red: no activity, yellow: may be active).

**Figure 2 antibiotics-14-00178-f002:**
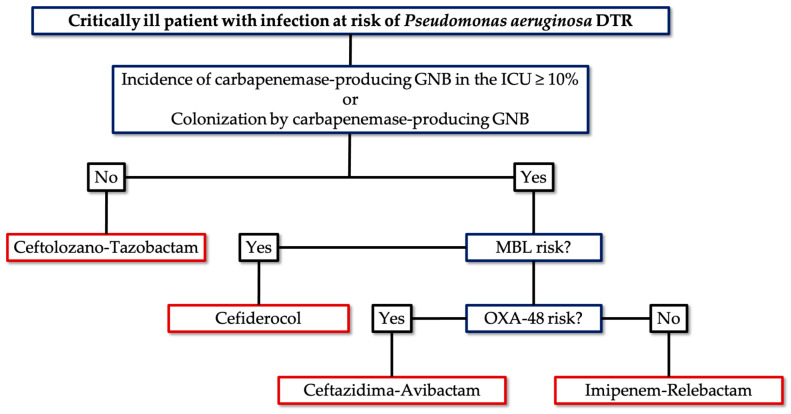
Empirical treatment algorithm for critically ill patients with infection at risk of difficult-to-treat *Peudomonas aeruginosa*.

**Table 1 antibiotics-14-00178-t001:** Classification of *P. aeruginosa* based on antimicrobial resistance (combining Magiorakos’s [43] and Kadri’s [44] definitions).

Concept	Definition
Multidrug-resistant *P. aeruginosa* (MDR-PA)	Non-susceptible to at least one antimicrobial agent in three or more antimicrobial classes active against *P. aeruginosa*.
Extensively drug-resistant *P. aeruginosa* (XDR-PA)	Non-susceptible to at least one agent in all but two or fewer antimicrobial classes.
Pandrug-resistant *P. aeruginosa* (PDR-PA)	Non-susceptible to any antimicrobial agent.
Difficult-to-treat *P. aeruginosa* (DTR-PA)	Non-susceptible to piperacillin–tazobactam, ceftazidime, cefepime, aztreonam, meropenem, imipenem–cilastatin, ciprofloxacin, and levofloxacin.

**Table 2 antibiotics-14-00178-t002:** Antibiotic dosage for the treatment of severe infections caused by DTR-PA.

Antibiotic	Dosage	pK/pD Target
Amikacin	25–30 mg/kg ^1^ (adjust according to levels)	C_max_/MIC ≥ 8–10 AUC_0–24h_/MIC 80–100
Aztreonam–Avibactam	0.5/0.167 g loading dose (30 min) 1.5/0.5 g (3 h infusion)/6 h	%*f*T > MIC
Cefepime–Enmetazobactam	2/0.5 g/8 h (2 h infusion)	*f*AUC_0–24h_/MIC
Cefepime–Taniborbactam	2/0.5 g/8 h (2–4 h infusion)	*f*AUC_0–24h_/MIC
Cefepime–Zidebactam	2/1 g/8 h	
Cefiderocol	2 g/8 h (2 g/6 h if GF > 120 mL/min) (3 h infusion)	%*f*T > 4 × MIC > 80%
Ceftazidime–Avibactam	2/0.5 g/8 h (2/0.5 g/6 h if GF > 130 mL/min) (3 h infusion)	100% *f*T > 4 × MIC
Ceftolozane–Tazobactam	1–2/0.5–1 g/ 8 h	100% *f*T > 4 × MIC
Colistin	9 MU (colistimethate), loading dose (0.5–1 h infusion) 4.5 MU/12 h (5.5 MU/12 h if GF ≥ 90 mL/min)	*f*AUC_0–24h_/MIC ≥ 12
Fosfomycin	8 g loading dose 16–24 g as a continuous infusion/24 h	100% *f*T > MIC
Imipenem–Relebactam	500/250 mg/6 h (30 min infusion)	*f*AUC_0–24h_/MIC
Meropenem–Vaborbactam	2/2 g/8 h (3 h infusion)	*f*AUC_0–24h_/MIC
Tobramycin	6–7 mg/kg ^1^ (adjust according to levels)	C_max_/MIC ≥ 8–10 AUC_0–24h_/MIC 80–100

AUC: area under the curve, C_max_: maximum concentration, MIC: Minimum Inhibitory Concentration, *f*: free fraction, T: time. ^1^ Adjusted body weight.

## Data Availability

Not applicable.

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
