# Peer review of "Difficult-to-Treat Pseudomonas aeruginosa Infections in Critically Ill Patients: A Comprehensive Review and Treatment Proposal"

_antibiotics, 2025, doi:10.3390/antibiotics14020178_

Round 1
Reviewer 1 Report
Comments and Suggestions for Authors
Reviewer’s comments
The review is comprehensive and provides valuable information and statistics; however, it could benefit from improved organization to enhance readability and clarity. The paragraphs could be more effectively grouped under relevant headings to create a stronger connection between the different pieces of information. Additionally, the numbering of the paragraph titles is somewhat confusing—simplifying or incorporating sub-headings would improve the structure. Furthermore, the frequent use of abbreviations throughout the manuscript may create confusion. The authors could consider using standard abbreviations for some terms while spelling others out in full to improve readability.
1. L35-36: The statement “however, exposure to an environment where PA is present for as little as 3–5 days can lead to colonization” appears to be a generalization. The referenced study specifically highlights that such colonization occurs under conditions of high exposure pressure, such as in intensive care units (ICUs). Please revise this statement accordingly.
2. The abbreviation “PA” is used throughout the manuscript to refer to Pseudomonas aeruginosa, but this is not in line with standard conventions. It would be more appropriate to use “P. aeruginosa” throughout the text, as this is the widely accepted and commonly used form.
3. L49-50: The citation for this statement seems to be incorrect. Please verify and correct the reference.
4. For clarity, please replace the abbreviation "NP" with the full term “nosocomial pneumonia,” as this would be more accessible to the reader.
5. L54: The statement “PA is its leading cause [12]” should be accompanied by the correct reference. Please ensure that all references throughout the manuscript are accurate and properly cited.
6. Figure 1: The color scheme in the figure, particularly the half-yellow/half-green and half-yellow/half-red sections, is somewhat unclear and not very descriptive. The authors should consider revising the figure to make it more intuitive and visually accessible for the reader.
7. The risk factors for P. aeruginosa (PA) and multidrug-resistant P. aeruginosa (MDR-PA) could be consolidated under a single heading, "Risk Factors," to streamline the content.
8. L136-138: The sentence in this section is quite unclear. The authors should revise this portion for better clarity and coherence.
Author Response
Comments 1: L35-36: The statement “however, exposure to an environment where PA is present for as little as 3–5 days can lead to colonization” appears to be a generalization. The referenced study specifically highlights that such colonization occurs under conditions of high exposure pressure, such as in intensive care units (ICUs). Please revise this statement accordingly.
Response 1: Thank you for the clarification; the sentence has been modified according to your suggestion.
Comments 2: The abbreviation “PA” is used throughout the manuscript to refer to Pseudomonas aeruginosa, but this is not in line with standard conventions. It would be more appropriate to use “P. aeruginosa” throughout the text, as this is the widely accepted and commonly used form.
Response 2: PA was used in the manuscript in order to make it more concise and less repetitive. In addition to P. aeruginosa, PA is used as an abbreviation (PA-MDR, PA-DTR...), but has been replaced by P. aeruginosa in the text according to your suggestion
Comments 3: L49-50: The citation for this statement seems to be incorrect. Please verify and correct the reference.
Response 3: The reference is:
- Vincent JL, Sakr Y, Singer M, Martin-Loeches I, Machado FR, Marshall JC, et al. Prevalence and Outcomes of Infection Among Patients in Intensive Care Units in 2017. JAMA. 2020 Apr 21;323[15]:1478–87.
In this article is stated: Among the 3540 patients who had gram-negative microorganisms identified on culture, the most common were Klebsiella species (973 patients [27%]), Escherichia coli (902 patients [25%]), Pseudomonas species (850 patients [24%]), and Acinetobacter species (602 patients [17%])
So, our sentence "Globally, P. aeruginosa is the third most frequently isolated GNB among patients hospitalized in Intensive Care Units (ICUs)" seems correct, according to the reference. Please, let us know if we are wrong.
Comments 4: For clarity, please replace the abbreviation "NP" with the full term “nosocomial pneumonia,” as this would be more accessible to the reader.
Response 4: NP has been replaced with nosocomial pneumonia. Thank you for your suggestion.
Comments 5: L54: The statement “PA is its leading cause [12]” should be accompanied by the correct reference. Please ensure that all references throughout the manuscript are accurate and properly cited.
Response 5: Thank you very much for your correction. We have re-writed the paragraph to be more accurate with the data.
Comments 6: Figure 1: The color scheme in the figure, particularly the half-yellow/half-green and half-yellow/half-red sections, is somewhat unclear and not very descriptive. The authors should consider revising the figure to make it more intuitive and visually accessible for the reader.
Response 6: I respectfully disagree, but if anything, the table is intuitive rather than precise, as it is not possible to provide exact data for each antibiotic against every resistance mechanism. I believe the traffic light color code is intuitive and allows for a general idea of which antibiotics can be considered first-line options for each mechanism and which ones should be used with caution.
Comments 7: The risk factors for P. aeruginosa (PA) and multidrug-resistant P. aeruginosa (MDR-PA) could be consolidated under a single heading, "Risk Factors," to streamline the content.
Response 7: Following your recommendation, we have unified the risk factors section. Thank you for your suggestion.
Comments 8: L136-138: The sentence in this section is quite unclear. The authors should revise this portion for better clarity and coherence.
Response 8: The sentence has been re-written in order to achieve better clarity following your suggestion.
We sincerely appreciate your review and suggestions and hope that this new version has reached sufficient quality to be published.
Reviewer 2 Report
Comments and Suggestions for Authors
The review addresses a highly relevant topic and presents it in an organized, clear manner, providing updated and necessary information. Below are some observations and comments that could improve the article:
-
It is necessary to correct the formatting of certain words. For example, the terms in vivo and in vitro should be italicized and written in lowercase.
-
Throughout the document, the pathogen's name is abbreviated in two different ways: P. aeruginosa and PA. It is important to standardize its abbreviation consistently across the text.
-
Table 1 presents the classification of P. aeruginosa based on resistance level. However, as currently displayed, it appears to be a single classification. It is necessary to clearly indicate that it consists of two separate classification proposals: one by Mariorakos et al. and the other by the DTR-PA classification.
-
In lines 196-197, it is stated that a risk factor for P. aeruginosa infection in the community is a C-reactive protein level <12.35 mg/dL. This statement is noteworthy and should be reviewed for accuracy. Please clarify the concept if necessary.
-
In several paragraphs, such as lines 401-405, the results of therapy against infections caused by Gram-negative bacilli are discussed, mentioning that P. aeruginosa is included among them. However, the specific results for this bacterium are not explicitly stated. Given that this review focuses on P. aeruginosa, these results should be clearly indicated.
Any comments
Author Response
Comments 1: It is necessary to correct the formatting of certain words. For example, the terms in vivo and in vitro should be italicized and written in lowercase.
Response 1: Thank you for your reminder. I have corrected that mistake
Comments 2. Throughout the document, the pathogen's name is abbreviated in two different ways: P. aeruginosa and PA. It is important to standardize its abbreviation consistently across the text.
Response 2. In accordance with your suggestion and that of Reviewer 1, P. aeruginosa has been used. However, we have retained "PA" in abbreviations such as DTR-PA or MDR-PA for linguistic economy.
Comments 3: Table 1 presents the classification of P. aeruginosa based on resistance level. However, as currently displayed, it appears to be a single classification. It is necessary to clearly indicate that it consists of two separate classification proposals: one by Mariorakos et al. and the other by the DTR-PA classification.
Response 3: Thank you for your suggestion. Table 1 title has been corrected following your recomendation
Comments 4: In lines 196-197, it is stated that a risk factor for P. aeruginosa infection in the community is a C-reactive protein level <12.35 mg/dL. This statement is noteworthy and should be reviewed for accuracy. Please clarify the concept if necessary.
Response 4: A C-reactive protein level <12.35 mg/dL has been identified as a independent risk factor for PA in CAP.But it´s important to highlight the last sentence of this section: "In general, risk factors and predictive scores show high sensitivity but low specificity, resulting in high negative predictive value but low positive predictive value. Therefore, it is crucial to understand local microbiology and resistance profiles to guide clinical decisions effectively"
Comments 5: In several paragraphs, such as lines 401-405, the results of therapy against infections caused by Gram-negative bacilli are discussed, mentioning that P. aeruginosa is included among them. However, the specific results for this bacterium are not explicitly stated. Given that this review focuses on P. aeruginosa, these results should be clearly indicated.
Response 5: Results focused on P. aeruginosa are indicated when available (in the main article or in supplementary material). Unfortunatly, due to low number of PA infections in this clinical trials, is frequent that PA results are not mentioned separately.
Thank you very much for your review and your contributions, I hope that the quality of the manuscript has improved enough to consider its publication.